# SlicerMorph photogrammetry: an open-source photogrammetry workflow for reconstructing 3D models

Oshane O. Thomas[1], Chi Zhang[2] and A. Murat Maga[1,3,*]

## ABSTRACT

High-fidelity three-dimensional (3D) models of skeletal specimens underpin many ecological and evolutionary analyses. Here we present a fully open pipeline inside the 3D Slicer platform that couples automatic image masking by the Segment Anything Model (SAM) with surface reconstruction by the OpenDroneMap (NodeODM) engine, all wrapped in a user-friendly extension. To test accuracy, we photographed 14 mountain-beaver skulls, reconstructed 3D models with the new pipeline and with our previous workflow and compared each model to its micro-CT reference using mean surface distance, root mean square error (RMSE), Hausdorff, and Chamfer metrics. Our improved pipeline that integrates masking to the model reconstructed lowered mean distance and RMSE by 10–15% across specimens and reduced visible artefacts around thin elements such as zygomatic arches; Hausdorff distance changed little, indicating that gains were global rather than confined to outliers. Our new extension provides a convenient workflow that integrates masking, scaling, and reconstruction under one interface.

KEY WORDS: Photogrammetry, OpenDroneMap, 3D morphometrics

## INTRODUCTION

Three-dimensional (3D) digitization has become a cornerstone in modern morphological research. It enables scientists to capture high-resolution data of skeletal remains, whole organisms, and other biological structures without physically disturbing the original specimens (Abhinav et al., 2021; Peng, 2008; Santana et al., 2019; da Silva et al., 2023). By generating digital replicas, researchers can dissect complex forms for comparative analyses in functional morphology, taxonomic classification, biomechanics, and evolutionary and developmental biology (Irschick et al., 2022; da Silva et al., 2023).

In particular, photogrammetry, which constructs a 3D surface model from multiple overlapping photographs taken at different positions, has experienced a surge in adoption due to its relatively low cost, portability, and ability to image objects of varied sizes. The resulting textured 3D models capture geometric and color information, facilitating various morphological and ecological interpretations. Furthermore, digitally archiving specimens in high-fidelity 3D form is crucial for globally distributed research collections, especially when dealing with fragile or rare organisms (Blagoderov et al., 2012; Vollmar et al., 2010).

Despite the growing popularity of photogrammetry, many existing workflows rely on proprietary software that can be prohibitively expensive or impose licensing restrictions. In parallel, the open-source community has generated numerous structure-from-motion (SfM) tools, but these tools often require advanced scripting or specialized computing environments and can be challenging to use if the researcher lacks the necessary technical background (Biegel, 2024; Patel et al., 2024; Vacca, 2019; Zhang and Maga, 2023).

Another challenge in many photogrammetry workflows is the need for background removal, which can be labor-intensive, depending on how it is done. Traditional approaches often require manual tracing or semi-automatic thresholding, introducing potential operator bias and risking the loss of fine structural details. Skulls with thin or fenestrated bony regions, such as the zygomatic arches of the skull, are especially susceptible to over-erosion during masking, resulting in incomplete or 'broken' geometries (Zhang and Maga, 2023).

In addition, artifact removal (e.g. bridging across suture lines or 'clogs' in narrow passages) can take significant time, limiting throughput for large sample sets. This is particularly problematic for studies that rely on subtle morphological signals to distinguish closely related groups. Dashti et al. (2022) suggest that discrepancies as small as 0.1–0.2 mm may obscure functional traits or early signals of morphological divergence in small mammals. These small-scale differences can reflect adaptive changes in dentition, cranial morphology, or muscle attachment sites.

Our previous study (Zhang and Maga, 2023) showed the possibility of consolidating various open-source tools into a workflow to generate 3D textured models from photographs using a low-cost, portable setup. This initial attempt was more focused on providing a detailed documentation of how to acquire photographs from skeletal specimens found in natural history collections for photogrammetry purposes. The requirement for manual background removal and semi-automatic masking that was only partially effective, resulted in less-than-ideal results for geometrically complex regions such as the pterygoid and zygomatic processes. Nonetheless, the study demonstrated the feasibility of using low-cost commodity hardware and open-source tools to acquire requisite photographs and build 3D textured models that are sufficiently accurate for most morphometric studies or teaching purposes.

Here, we improve on this study by integrating the Segment Anything Model (SAM) (Kirillov et al., 2023 preprint) with the structure-from-motion reconstruction [NodeODM, part of the OpenDroneMap project (Patel et al., 2024)] and combining these tools within the 3D Slicer (Fedorov et al., 2012) environment

[1]Center for Development Biology and Regenerative Medicine, Seattle Children's Research Institute, Seattle, Washington, 98101, United States of America. [2]Department of Biomedical Sciences, Texas A&M University College of Dentistry, Dallas, Texas, 98101, USA. [3]Division of Craniofacial Medicine, Department of Pediatrics, University of Washington, Seattle, Washington, 98105, USA.

*Author for correspondence (maga@uw.edu)

A.M.M., 0000-0002-7921-9018

Biology Open

to unify data acquisition, segmentation, reconstruction, and final model generation. Our updated workflow aims to streamline background masking while improving the quality of the obtained mask, and explore new parameters introduced to ODM since the publication of the previous study to improve 3D model generation. Resultant textured models can be imported into the SlicerMorph

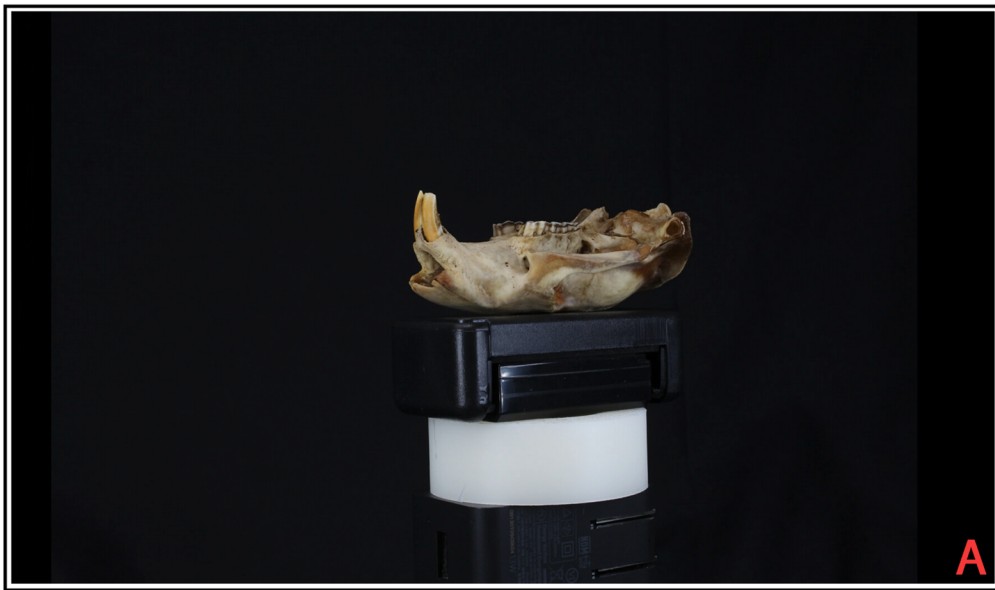

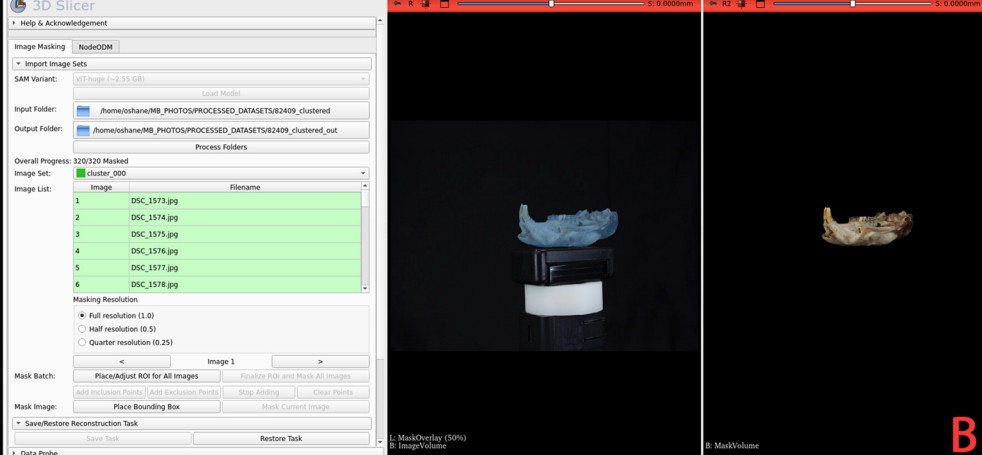

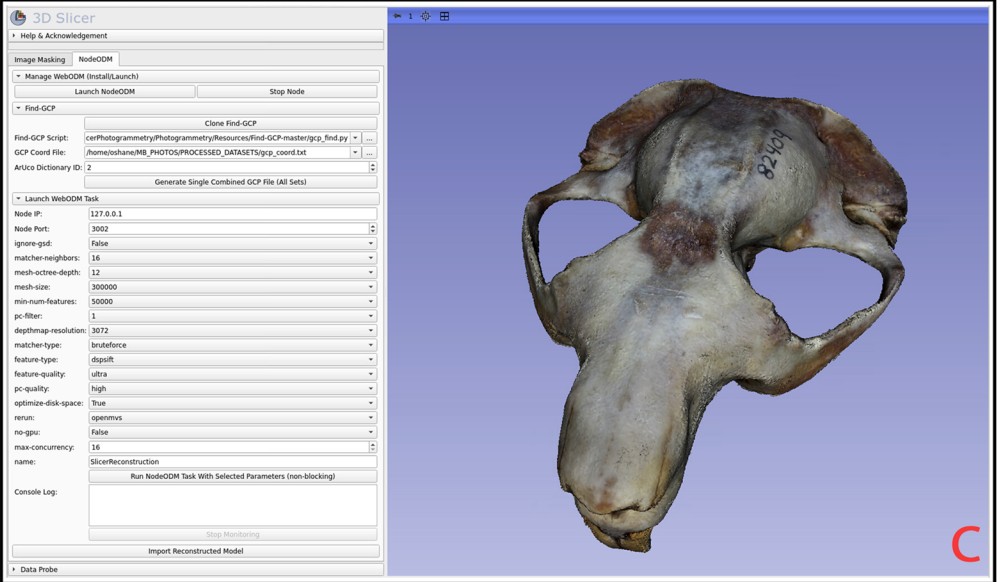

**Fig. 1. Overview of the photogrammetry extension workflow using a mountain beaver skull dataset.** (A) Example of one original photograph from the 320-image dataset, which was taken as the specimen mounted on a turntable. The photogrammetry user interface consists of two module tabs (Image Masking and NodeODM) that focus on different aspects of the workflow. (B) The Image Masking interface displaying a fully masked dataset. The left 2D view shows the original photograph overlaid with a blue mask generated using the SAM, highlighting the segmented specimen. The right 2D view presents the same image post-segmentation, with the background removed, isolating the skull. (C) ODM interface post-photogrammetry reconstruction with default parameters set in the Find GCP and Launch NodeODM Task subsections. The resulting reconstructed and textured 3D skull model is displayed imported directly into the 3D viewport via the 'Import Reconstructed Model' functionality.

**Table 1. Reconstruction errors for two photogrammetry workflows (Zhang et al. 2023 versus the updated workflow) compared to micro-CT references**

| Specimen ID | Workflow/method | Average distance | RMSE | Hausdorff | Chamfer |
|---|---|---|---|---|---|
| 39867 | Zhang et al., 2023 | 0.378875 | 0.491515 | 4.118857 | 1.567254 |
| | Current study | 0.319101 | 0.419799 | 4.175258 | 1.455372 |
| 82710 | Zhang et al., 2023 | 0.414554 | 0.559897 | 4.728765 | 1.568789 |
| | Current study | 0.350505 | 0.496255 | 4.526612 | 1.453461 |
| 34050 | Zhang et al., 2023 | 0.385806 | 0.503629 | 3.983145 | 1.728082 |
| | Current study | 0.32047 | 0.420359 | 4.090909 | 1.588246 |
| beaver_0 | Zhang et al., 2023 | 0.377915 | 0.526913 | 4.433732 | 1.481809 |
| | Current study | 0.341453 | 0.507385 | 4.385388 | 1.402892 |
| 82409 | Zhang et al., 2023 | 0.367641 | 0.485357 | 4.606842 | 1.624009 |
| | Current study | 0.336306 | 0.460284 | 4.580018 | 1.531256 |
| 31870 | Zhang et al., 2023 | 0.393741 | 0.531894 | 4.190532 | 1.611603 |
| | Current study | 0.334341 | 0.440317 | 4.305987 | 1.48024 |
| 79564 | Zhang et al., 2023 | 0.385528 | 0.524074 | 4.403106 | 1.647713 |
| | Current study | 0.35191 | 0.49079 | 4.299145 | 1.564994 |
| 34072 | Zhang et al., 2023 | 0.379874 | 0.495752 | 4.003932 | 1.566099 |
| | Current study | 0.33907 | 0.455136 | 4.103916 | 1.546559 |
| 34073 | Zhang et al., 2023 | 0.361263 | 0.472148 | 4.37394 | 1.5581 |
| | Current study | 0.334394 | 0.456411 | 4.36975 | 1.536601 |
| 35998 | Zhang et al., 2023 | 0.432197 | 0.574538 | 4.082794 | 1.640273 |
| | Current study | 0.380621 | 0.501963 | 3.868277 | 1.524142 |
| 39384 | Zhang et al., 2023 | 0.362147 | 0.470161 | 3.892134 | 1.552798 |
| | Current study | 0.356324 | 0.463967 | 3.936572 | 1.520624 |
| 34083 | Zhang et al., 2023 | 0.362919 | 0.496577 | 4.348915 | 1.577649 |
| | Current study | 0.312413 | 0.426636 | 4.525327 | 1.445302 |
| 39385 | Zhang et al., 2023 | 0.381905 | 0.507052 | 3.938614 | 1.585428 |
| | Current study | 0.347265 | 0.46862 | 3.949466 | 1.51819 |
| 34081 | Zhang et al., 2023 | 0.394414 | 0.532447 | 4.425598 | 1.510912 |
| | Current study | 0.354806 | 0.499825 | 4.440951 | 1.497483 |

Models were aligned to the CT via ICP, then uniformly sampled to measure Euclidean (L2) distances, RMSE (emphasizing outliers), Hausdorff distance (worst local discrepancy), and Chamfer distance (bidirectional mismatch). Across specimens, the new workflow generally yields lower average and RMSE values, with Hausdorff and Chamfer distances remaining comparable or improved. 'UWBM' indicates specimens from the University of Washington Burke Museum; 'beaver0' is from a private collection.

extension allows researchers to seamlessly continue their workflow, whether for visualization, landmarking, or other morphometric analysis (Rolfe et al., 2021) (Fig. 1).

In this study, we quantitatively assess the improvements to the geometric accuracy of the models reconstructed from the photogrammetry extension by comparing the models generated

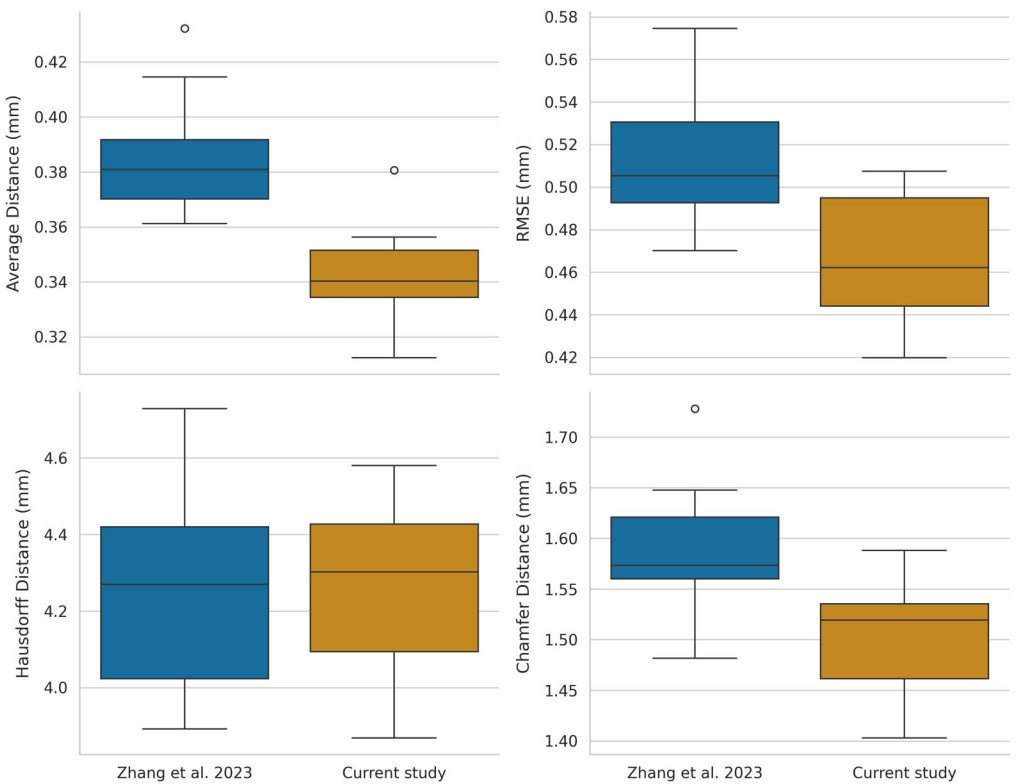

**Fig. 2. Box plot comparison of reconstruction errors from Zhang et al. (2023) versus our updated photogrammetry workflow, using micro-CT data as the ground–truth reference.** Each panel shows one of four error metrics – average distance, RMSE, Hausdorff distance, and Chamfer distance – across multiple rodent skull specimens. The updated workflow yields consistently lower average distance and RMSE values (10–15%) in most specimens, indicating improved overall fidelity in delicate cranial regions. Although Hausdorff and Chamfer distances vary by specimen – occasionally reflecting incomplete photo coverage – the overall trend suggests the new pipeline provides more accurate and reproducible 3D models for ecological and evolutionary research.

from both workflows to their ground-truth model acquired by high-resolution micro-CT. We provide quantitative (mean distance, RMSE, Hausdorff, Chamfer) and qualitative (visual) evidence of improved capture of intricate skeletal elements.

## RESULTS AND DISCUSSION
### Overview of model quality and accuracy
Across the *Aplodontia rufa* skull dataset, the updated workflow systematically yielded more accurate 3D models than the previous pipeline (Zhang and Maga, 2023). Across 14 specimens, we observed a ∼10–15% decrease in mean surface error relative to the micro-CT reference (Table 1, Fig. 2). This improvement was particularly evident in delicate bony structures (e.g. zygomatic arches, orbital rims), where the older approach often generated small artifacts or over-smoothed regions. By contrast, the new method's integration of the SAM and refined ODM parameters produced smoother surfaces with fewer extraneous polygons.

Alongside these quantitative gains in average distance and root mean square error (RMSE), we noted qualitatively higher fidelity (Fig. 3). Zygomatic arches and foramina, often prone to background noise or partial mis-segmentation in the previous study, were reconstructed with fewer breaks or bridging polygons. Although Hausdorff and Chamfer distances sometimes varied by specimen, chiefly when portions of the skull were incompletely photographed, global mean error and RMSE improvements indicate a more consistent capture of external morphology.

### Comparisons between old and new workflows
Compared to the previous study, the updated pipeline consistently reduced unidirectional discrepancy from experimental (E) reconstructed models to ground-truth (G) models (E→G) by 0.04–0.07 mm, allowing more detailed capture of delicate bone edges. For example, in specimen UWBM 82710, the average error dropped from 0.4146 mm to 0.3505 mm, while RMSE fell from 0.5599 mm to 0.4963 mm. These gains are not merely technical but translate into fewer hours of manual editing since morphological landmarks now appear more sharply defined. Although Hausdorff's distance occasionally increased slightly (e.g. from ∼4.37 mm to ∼4.53 mm) in some specimens, this typically reflected localized deficits, such as reflective areas or thin edges missed by the camera. Because ecological analyses often prioritize overall shape integrity over single outliers, the strong improvements in mean and RMSE are likely of greater practical importance.

Visual assessments reinforce these numeric trends. Due to suboptimal image masking, the prior pipeline occasionally 'clogged' the pterygoid or palatal regions with noisy vertices. In contrast, current approach better delineated these structures (Fig. 3). Warmer colors in deviation maps remain largely restricted to the periphery, indicating that interior cranial regions are aligned well with micro-CT references. The consistent preservation of fine anatomical details, especially around delicate areas such as the zygomatic arches and orbital margins, demonstrates the improved fidelity and reduced artifacts achieved by our updated photogrammetry workflow (Fig. 4).

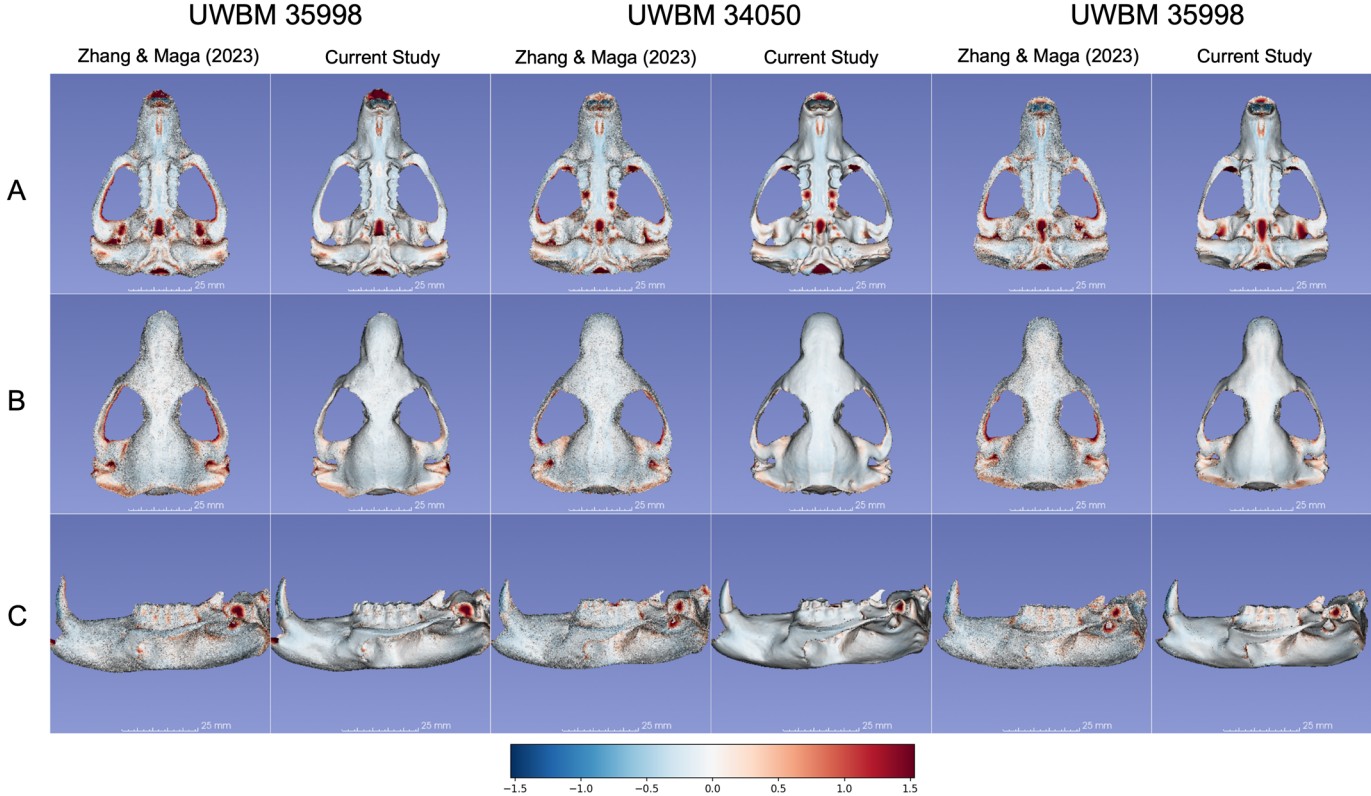

**Fig. 3. Side-by-side photogrammetric reconstructions of three *A. rufa* (mountain beaver) skull specimens – UWBM 39867 (top two rows), UWBM 82710 (middle two rows), and UWBM 34050 (bottom two rows) – shown in A (dorsal view), B (ventral view), and C (oblique posterior–lateral view).** For each specimen and orientation, the upper image is derived from the Zhang and Maga (2023) workflow, and the lower image is reconstructed via the photogrammetry extension being presented. Warmer colors in the distance maps (bottom scale in mm) indicate larger deviations, whereas cooler colors signify closer agreement. Across all three specimens, the new workflow yields fewer triangular artifacts around the orbital and zygomatic boundaries avoids clogging critical foramina and reduces over-smoothed patches in under-photographed areas. Its enhanced SAM-based masking more accurately separates the specimen from the background, resulting in consistently improved model fidelity and more anatomically faithful reconstructions.

| UWBM 34050 | UWBM 39867 | UWBM 82710 | UWBM 34069 |
|---|---|---|---|

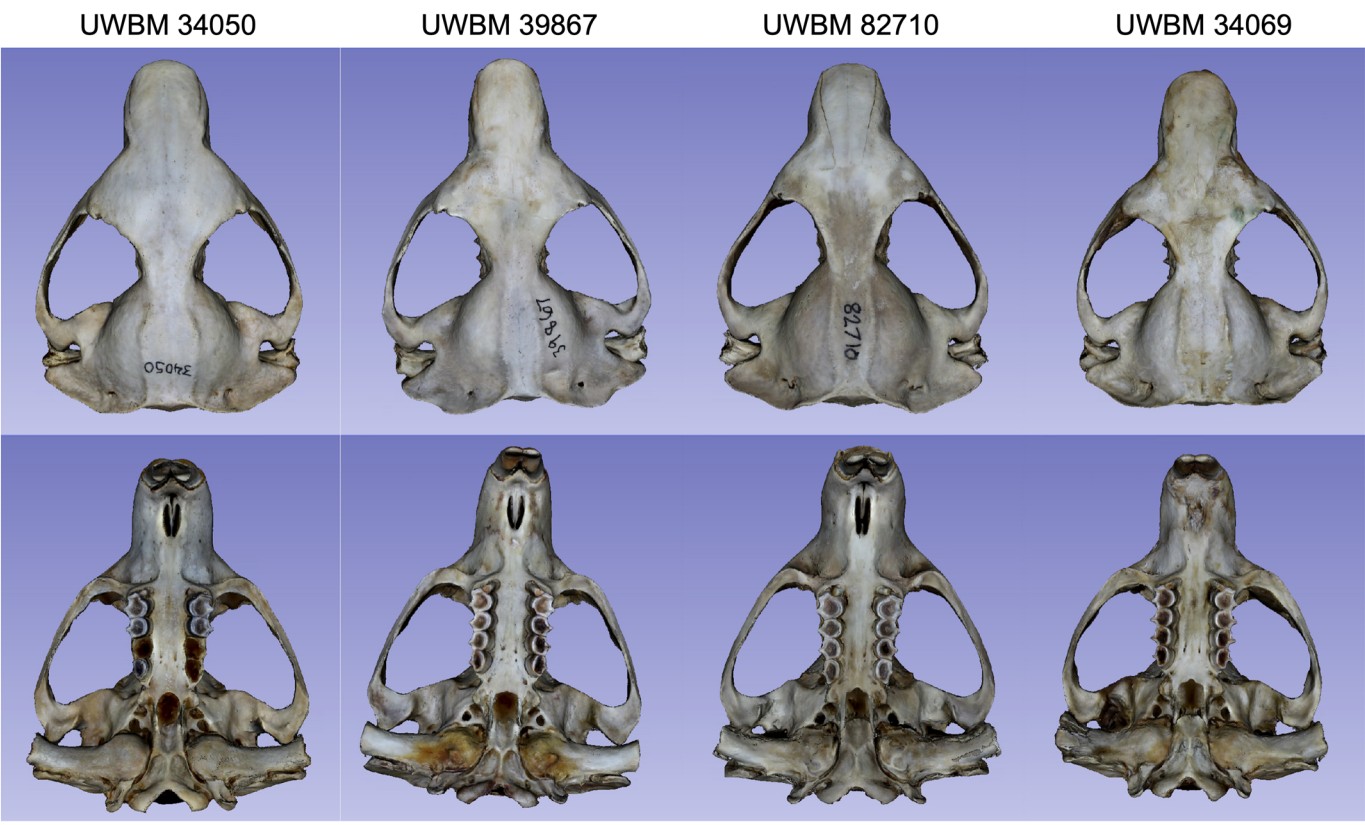

**Fig. 4. Dorsal (left columns) and ventral (right columns) views of four mountain beaver (*A. rufa*) skull specimens reconstructed using the photogrammetry extension described in this study.** Specimen IDs from top to bottom: UWBM 34069, UWBM 82710, UWBM 39867, and UWBM 34050. The reconstructed models demonstrate minimal artifacts and well-defined cranial features, particularly around delicate anatomical structures such as the zygomatic arches, orbital margins, and palatal foramina, highlighting the fidelity achievable with the streamlined, open-source photogrammetry workflow.

### Metrics and their interpretations

To characterize each reconstruction's geometric accuracy, we assessed four principal error metrics: average (mean) distance from E→G, RMSE, Hausdorff distance, and Chamfer distance (Table 1). Mean distance and RMSE collectively illustrate the model's overall fidelity. A low mean distance (∼0.32 mm) can confirm that smaller external details (zygomatic arches, sutures) are faithfully captured, assuming this value remains well within the natural morphological variation for a skull measuring 30–40 mm in length. RMSE places extra weight on larger discrepancies; thus, if we see a rise in RMSE (e.g. from 0.40 mm to 0.50 mm), it indicates that a small region of the skull may be poorly reconstructed, even if the rest is accurate. The presence of 'patched' surfaces in highly fenestrated bones underscores the need for thorough photo coverage.

Hausdorff distance identifies the worst-case discrepancy, often localized around missed camera angles or reflective edges. While a moderate rise in Hausdorff suggests incomplete imaging, the effect on ecological interpretations is less dire when the overall mean and RMSE are substantially reduced. Chamfer distance, a bidirectional average, usually aligns with mean error but also captures unmatched geometry. In our case, internal cranial structures present in CT but absent in photogrammetric data inflate G→E (CT→external surface) errors, highlighting the intrinsic mismatch of internal versus external anatomy. Researchers focusing on external shape typically emphasize E→G, ensuring a more direct measure of how faithfully the external model reproduces the micro-CT's outer surface.

### Utility for ecological and evolutionary applications

Our approach supports various morphological studies by reducing reconstruction errors in structurally complex regions. For instance, subtle shape differences on the mountain beaver skull, such as small alveolar expansions, ridge formation, or cranial curvature, become more readily detectable. This allows researchers to investigate diet-induced shape variation, population divergence, or interspecific comparisons without extensive manual correction. High-fidelity models also facilitate geometric morphometric analyses (e.g. dense or semi-landmarks) that rely on consistent surface detail.

While the new workflow mitigates many typical photogrammetry hurdles, high-quality photography remains a crucial part of the full pipeline. In most cases, 3D geometry of the morphologically complex regions (e.g. zygomatic arches or pterygoid processes) can be further improved by acquiring more photograms of these regions in different angles. For a more detailed treatment on best practices of photography for photogrammetry purposes, we refer the reader to our previous study (Zhang and Maga, 2023). In summary, we strongly advise dense angular overlap between pictures; acquiring 60–80% overlap and multiple vertical angles ensures all structures appear in at least two images.

### Conclusions

Our findings demonstrate that integrating the SAM with NodeODM in 3D Slicer notably improves photogrammetric reconstructions of *A. rufa* skulls compared to the prior workflow (Zhang and Maga, 2023). Automating background masking and systematically

optimizing parameters yielded a 10–15% reduction in mean distance and RMSE, which is particularly beneficial for thin cranial regions. These accuracy gains minimize laborious edits and enhance micro-CT alignment, making subtle morphological signals more discernible. It should be noted that further geometric accuracy can be improved by taking additional photographs of the problematic regions in orientations not acquired originally and supplementing the existing dataset.

Beyond lowering error rates, the updated pipeline's open-source design supports broader taxonomic and collection management applications. By embedding image masking, reconstruction, and morphometric tools within one ecosystem, researchers can efficiently generate high-fidelity 3D models, catalyzing new studies or outreach activities in functional morphology, and evolutionary biology using an open-source platform.

## MATERIALS AND METHODS

### Overview of the extension and workflow

The workflow begins with importing photographs (e.g. skulls) into the photogrammetry module, which leverages SAM to isolate specimens from image backgrounds rapidly. Masking can be performed batch-wise or refined on an individual-image basis, significantly reducing the manual effort typically required. Following image processing, the extension directly integrates with NodeODM to reconstruct high-resolution 3D models.

### Software architecture and dependencies

Photogrammetry was developed for 3D Slicer (version 5.9; Fedorov et al., 2012) and is currently available through the 3D Slicer extension catalogue only for the Linux operating system. It depends on two primary Slicer extensions – PyTorch and SlicerMorph – as well as several Python libraries, which are automatically installed by the extension the first time it is launched:

- segment-anything for deep-learning-based image masking (Kirillov et al., 2023 preprint).
- pyodm for interacting with OpenDroneMap's NodeODM (Patel et al., 2024).
- OpenCV and Pillow for image processing and I/O.

NodeODM reconstruction requires Docker and, optionally, the NVIDIA Container Toolkit for GPU acceleration for NVIDIA GPUs. For users who do not want to deal with the technical aspects of setting up containers, or users on Windows and Mac operating systems who would like to benefit from this extension, we recommend using the free pre-configured cloud instances that are available freely from the MorphoCloud project. These pre-configured cloud instances provide a GPU-equipped, turnkey, powerful cloud environment with dozens of cores and over 100GB of RAM that enables users to process thousands of photographs without hitting any hardware restrictions.

### Installation and setup

We describe two primary installation options for the photogrammetry Extension for 3D Slicer:

- MorphoCloud on demand (recommended):
- Users access the MorphoCloud platform (https://github.com/MorphoCloud), a GPU-enabled environment preconfigured with all required software (Docker, NodeODM, NVIDIA Container Toolkit, PyTorch).
- Launch the Photogrammetry module directly from the SlicerMorph submenu of the 3D Slicer's Module Selector to begin processing immediately.
- Local installation on linux:
- Install Docker, NVIDIA GPU drivers, and the NVIDIA Container Toolkit if GPU acceleration is desired.
- Confirm Docker functionality (e.g. docker run –rm –gpus all nvidia/cuda:11.8.0-base).
- Install 3D Slicer (version 5.8) and use the Extension Manager (Linux) to install "SlicerMorph Photogrammetry. " PyTorch and SlicerMorph will be automatically installed as dependencies.

- On Windows and macOS, users must manually clone and load the repository as a scripted module (more detailed instructions are provided online).

### Image acquisition and masking

While robust to various image acquisition methods, we recommend consistent illumination and minimal background clutter for optimal results, following the recommendations in Zhang and Maga (2023). Images can be captured using a basic DSLR camera on a programmable turntable within a diffused lighting setup. For effective reconstruction, photographs should have approximately 70–80% overlap.

Once imported into the extension, the images are masked using SAM. Users select the masking resolution (full, half, or quarter) depending on hardware capability. Low-resolution masking is computationally less taxing and faster at the expense of accuracy. Two masking methods are available:

- Batch masking: A bounding box is placed around the specimen in one representative image, propagating automatically to all images in a predefined set.
- Single image masking: Individual adjustments are made to refine challenging images, including placing inclusion/exclusion markers for precision.

Masked images and binary masks, preserving essential camera metadata, are automatically stored for subsequent 3D reconstruction steps.

### Photogrammetry reconstruction via NodeODM

After image masking, users reconstruct 3D meshes through NodeODM, accessible directly from the photogrammetry extension.

### Optional scaling (via ArUco Markers)

Users may include ArUco markers in images to provide accurate physical scaling of the specimen. The extension automates the creation of Ground Control Point (GCP) files ("Find-GCP"), allowing NodeODM to incorporate scale and position accuracy into the final model. For instructions on creating a local coordinate system using Aruco markers, see Section 2.3 of the supplemental online material from Zhang et al. (2023).

### Launching NodeODM

The module provides a simple interface for launching a local NodeODM container via Docker or connecting to an existing remote server. The user should provide the IP address and the port number NodeODM is running for the latter.

### Reconstruction parameters

The module offers recommended settings, balancing detail with computational efficiency for reconstructing on the pre-configured cloud instance (*blinded*) instances for most morphological applications:

- mesh-octree-depth=12
- mesh-size=300,000–500,000
- feature-quality=ultra
- pc-quality=high
- no-gpu=false (recommended for stable texturing)

Users initiate reconstruction via the "Run NodeODM Task" button, and the module provides real-time updates.

### Model retrieval

Completed 3D models (OBJ and associated texture images) are automatically downloaded, organized, and available for direct import into 3D Slicer for immediate visualization and measurement. The reconstruction parameters and configuration details are saved as JSON files to facilitate reproducibility.

### Worked example: mountain beaver skull (UWBM 82409)

To illustrate the workflow, we reconstructed a mountain beaver (*A. rufa*) skull (specimen UWBM 82409) from 320 images. Following Zhang and Maga (2023), images were taken to ensure high overlap (~70–80%) and consistent illumination. After batch-masking the images in about 20 min,

**Table 2. Top-performing Taguchi L16 configuration for NodeODM reconstruction and unidirectional error metrics relative to micro-CT scans**

| Parameter/metric | Value |
| --- | --- |
| ignore-gsd | FALSE |
| matcher-neighbors | 16 |
| mesh-octree-depth | 12 |
| mesh-size | 300 000 |
| min-num-features | 50,000 |
| pc-filter | 2 |
| feature-quality | ultra |
| pc-quality | ultra |
| Mean Distance_E→G | 0.300535 |
| RMSE_E→G | 0.402543 |
| Hausdorff_E→G | 3.918398 |
| Standard Deviation_E→G | 0.267805 |
| Mean Distance_G→E | 1.064301 |
| RMSE_G→E | 1.550204 |
| Hausdorff_G→E | 8.31234 |
| Standard Deviation_G→E | 1.127119 |
| Chamfer | 1.364836 |

All distances are in mm.

'E→G' indicates distances from experimental reconstruction to ground truth; 'G→E' indicates the reverse. Differences in E→G and G→E arise when the micro-CT includes internal geometry not captured by surface-based photogrammetry, thereby inflating the G→E distances when the reference points lack corresponding structures in the reconstruction.

we refined select images manually, for which automatic removal of the mounting platform and clay had failed.

NodeODM is launched locally (GPU enabled) with the default parameters described above. The user optionally provides a GCP list for the correct physical scaling of the specimen. Reconstruction yields a detailed, textured mesh that is automatically imported back into a 3D Slicer for evaluation. The images, masked outputs, and final reconstructed models are publicly available at https://app.box.com/shared/static/z8pypqqmel8pv4mp5k01phil frqep8xm.zip.

A graphical overview of this example is presented in Fig. 1, illustrating the masking and reconstruction pipeline. The code for our module can be found at https://github.com/SlicerMorph/SlicerPhotogrammetry.

## Optimizing photogrammetry parameters via taguchi design

We determined the optimal NodeODM reconstruction settings for the *A. rufa* skulls dataset through a Taguchi L16 design (Montgomery, 2017), varying seven major parameters such as mesh-octree-depth, mesh-size, and ignore-gsd while holding other baseline factors constant (e.g. orthophoto-resolution, texturing-single-material). We programmatically submitted masked images (plus optional ground-control points) to a local NodeODM instance using the Python pyodm library to facilitate this process. This approach systematically enumerated 16 unique parameter combinations (e.g. matcher-neighbors=0 or 16, pc-filter=1, 2, or 3) and recorded the resulting textured models.

Because slight misalignments can persist, we first applied a three-point registration to each reconstructed mesh to approximate alignment with a micro-CT 'gold standard' scan. We then refined each alignment via Iterative Closest Point (ICP) using Open3D (Zhou et al., 2018 preprint). It should be noted that these alignments are 'rigid' in nature and have no effect on the geometry of the reconstructed models; they simply alter the model's position in 3D space. Reconstructed and reference meshes were uniformly sampled to point clouds (260,000 points) for distance-based error calculations (mean, RMSE, Hausdorff, and standard deviation). Finally, we ranked parameter combinations by mean distance from reconstruction to micro-CT (lowest is best). This Taguchi-based methodology allowed us to evaluate various parameters efficiently while balancing runtime and resource considerations. The best configuration – listed in Table 2 – provided minimal geometric error across multiple metrics.

## Acknowledgements

We would like to thank OpenDroneMap developers for providing technical support throughout this project and making their software freely available for everyone. Micro-CT scanning was performed at the SCRI MicroCT Imaging Facility (RRID: SCR 024678) using a Bruker SkyScan 1272 micro-CT system purchased with an NIH Shared Instrumentation Grant (S10OD032302).

## Competing interests

The authors declare no competing or financial interests.

## Author contributions

Conceptualization: A.M.M.; Data curation: C.Z.; Formal analysis: O.O.T.; Funding acquisition: A.M.M.; Methodology: O.O.T.; Software: O.O.T.; Supervision: A.M.M.; Writing – original draft: O.O.T.; Writing – review & editing: O.O.T., C.Z., A.M.M.

## Funding

Development of the Photogrammetry extension was supported by grants DBI/2301405 and OAC/2118240 from the National Science Foundation (NSF) awarded to A.M.M. at Seattle Children's Research Institute. Additional funding for parts of this research was provided by NSF OAC/2118240 (HDR Imageomics Institute) and ACCESS [BIO180006] (MorphoCloud). Open Access funding provided by University of Washington. Deposited in PMC for immediate release.

## Data and resource availability

The code, scripts, and user documentation for the photogrammetry extension are hosted at https://github.com/SlicerMorph/SlicerPhotogrammetry. The example dataset (specimen UWBM 82409) is provided at https://seattlechildrens1.box.com/v/PhotogrammetrySampleData and is publicly available on OSF [or a similar repository] under a permanent DOI. The repository also includes complete pipeline configurations (JSON files), enabling users to replicate our procedures fully. Other supporting data can be found in the supplementary information.

## Peer review history

The peer review history is available online at https://journals.biologists.com/bio/lookup/doi/10.1242/bio.062126.reviewer-comments.pdf

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
