## [Peer Review File · Biology Open]

SlicerMorph Photogrammetry: An Open-Source Photogrammetry Workflow for Reconstructing 3D Models

Oshane O. Thomas, Chi Zhang and A. Murat Maga

DOI: 10.1242/bio.062126

Editor: Lewis Halsey

Review timeline

Original submission: 22 June 2025

Editorial decision: 30 June 2025

First revision received: 21 July 2025

Accepted: 25 July 2025

Original submission

First decision letter

MS ID#: bio.062126

MS Title: SlicerMorph Photogrammetry: An Open-Source Photogrammetry Workflow for Reconstructing 3D Models

Authors: Oshane O. Thomas, Chi Zhang and A. Murat Maga

I have now reached a decision on the above manuscript.

The reviewer reports are shown at the bottom of this email or can be accessed, together with a copy of this decision letter, by going to:

As you will see, the reviewers raised a number of substantial criticisms that prevent me from accepting the paper at this stage.

They suggest, however, that a revised version might prove acceptable, if you can address their concerns. If you think that you can deal satisfactorily with the criticisms on revision, I would be pleased to see a revised manuscript. We would then return it to the reviewers.

At this stage, we also ask you to ensure your manuscript complies with our formatting guidelines. Provided you are able to fully address the referees' comments, we are positive about publication of your paper (we accept over 95% of revision submissions) and therefore hope you won't mind any extra work involved in reformatting your manuscript at this point.

Please ensure that you clearly highlight all changes made in the revised manuscript. Please avoid using 'Tracked changes' in Word files as these are lost in PDF conversion.

I should be grateful if you would also provide a point-by-point response detailing how you have dealt with the points raised by the reviewers in the 'Response to Reviewers' box. Please attend to all of the reviewers' comments. If you do not agree with any of their criticisms or suggestions please explain clearly why this is so.

Reviewer 1

Comments for the author

This manuscript presents a well-structured open-source pipeline for biological photogrammetry within 3D Slicer. The approach is technically sound and addresses important usability and reproducibility challenges in morphological workflows.

Please refer to the attached document for the full review, including detailed comments and suggested revisions organized according to Biology Open's evaluation criteria.

“SlicerMorph Photogrammetry: An Open and Reproducible Photogrammetry Workflow Integrated into 3D Slicer”

The authors present a new open-source photogrammetry pipeline integrated into the 3D Slicer environment, which combines the Segment Anything Model (SAM) for automated image masking with NodeODM for surface reconstruction. The method is implemented as a user-friendly module and validated using 14 mountain beaver skulls, with micro-CT scans as the ground truth. The authors report quantitative improvements of 10-15% in mean distance and RMSE over a previous version of their workflow, along with improved visual fidelity in delicate anatomical structures. The tool is publicly available and designed to facilitate accessible and high-quality 3D model generation for morphometric and ecological research.

Experimental Quality

The manuscript provides a thorough comparison of the new SlicerMorph Photogrammetry workflow with a previous implementation, using multiple quantitative metrics (mean surface distance, RMSE, Hausdorff, Chamfer) benchmarked against micro-CT references. This use of micro-CT as a “ground truth” serves as an appropriate control for validating reconstruction accuracy.

The methods employed—including deep-learning-based masking (Segment Anything), structure-from-motion (OpenDroneMap), and quantitative assessment using ICP alignment and distance metrics—are appropriate for the question at hand. The workflow is well matched to the task of reconstructing high-fidelity 3D models of biological specimens.

The authors support their findings with both visual comparisons and a comprehensive Table reporting reconstruction metrics (mean distance, RMSE, Hausdorff, Chamfer) across all 14 specimens. This table provides excellent transparency and allows readers to assess variability in performance across samples.

Although no formal statistical hypothesis testing is conducted, the authors use relevant and interpretable performance metrics to support their claims. Given the Methods & Techniques nature of the paper, the approach to analysis is sound and appropriate.

Revision requested:

The authors may consider including a signed distance metric (e.g., via signed point-to-surface distances or scalar fields) in addition to the unsigned metrics (RMSE, Hausdorff, Chamfer). Signed distance maps not only quantify systematic over- or under-estimation in specific anatomical regions but also provide a clear visual interpretation of where surfaces are over- or under-reconstructed relative to the ground truth, which would complement the existing evaluation.

Reproducibility

The manuscript provides a thorough and clearly organized description of the photogrammetry pipeline, including installation instructions, system requirements, parameter settings, and execution steps. These are complemented by a well-documented GitHub repository (SlicerMorph/SlicerPhotogrammetry) that includes the full source code, module documentation, and configuration files.

The tool is currently best supported on Linux, where the extension can be installed directly through the 3D Slicer Extension Manager. For Windows and macOS users, a more complex manual setup is required, which may limit immediate usability for some researchers. The authors mitigate this by offering a cloud-based alternative (MorphoCloud), which provides an excellent workaround, though it does depend on third-party hosting. Clarifying long-term support plans for non-Linux users or containerized versions would be beneficial.

The authors also provide a public dataset (UWBM 82409), which includes the photographs used for reconstruction, but does not currently include the corresponding masks, 3D models, or error metrics. While the availability of raw input data is helpful, reproducibility would be greatly improved if at least one full example output were provided—including the generated mesh, masking files, and evaluation results.

More broadly, sharing even a subset of additional reconstructed specimens from the 14-sample validation would further enhance transparency and facilitate independent assessment of generalizability.

Overall, the methods are well described and the open-source infrastructure is robust, but expanding the scope of shared data and improving accessibility for non-Linux users would strengthen reproducibility.

Revisions requested:

- Please consider completing the example dataset (UWBM 82409) by adding the associated masks, reconstructed model, and evaluation output files (e.g., ICP-aligned mesh, deviation maps, metric results).
- Sharing at least 1-2 more specimens, even just the final reconstructions with configuration files, would help demonstrate consistency and further support reproducibility.
- Consider clarifying the long-term availability and platform support for users on Windows/macOS who cannot use MorphoCloud or prefer local workflows.

Completeness

The conclusions of the manuscript are well supported by the data. The observed improvements in geometric accuracy (as assessed by mean surface error and RMSE) align with both the visual and quantitative results presented. The authors interpret their results appropriately, without overstatement, and explicitly discuss cases where improvements are smaller or absent (e.g., Hausdorff distance increases due to localized imaging gaps).

The experimental design is sound. The sample size ($n = 14$ skulls) is sufficient for method validation, and the consistent imaging and reconstruction protocols are appropriate for comparing workflows.

Importantly, the authors acknowledge the inherent limitations of photogrammetry compared to CT—particularly in internal structures—and note how image quality and coverage can impact reconstruction accuracy. They also refer readers to their prior study for complementary best practices in photographic setup.

Overall, the manuscript presents a complete and transparent evaluation of the method's capabilities and limitations.

Revision requested:

The authors could consider highlighting the limitations more explicitly in the conclusions section—for example, reminding readers that the pipeline is optimized for external morphology and still depends on high-quality photographic input.

Scholarship

The manuscript is generally well contextualized within the relevant literature. The authors reference key technologies, prior photogrammetry workflows, and practical considerations in 3D model generation and morphometrics. They build directly on their previous work and cite recent open-source initiatives and comparative studies in the field.

While the discussion focuses primarily on improvements over their earlier method, the manuscript would benefit from broader critical engagement with alternative 3D reconstruction approaches (e.g., structured light scanning, laser scanning). Including references that address known limitations or failure modes of photogrammetry would provide a more balanced perspective and help delineate the specific contexts where this tool excels.

Nonetheless, the current framing is suitable for a Methods & Techniques article and reflects a solid understanding of both the technical and biological domains.

Although the validation using 14 mountain beaver skulls is appropriate, a brief note on the method's potential generalizability to other specimen types would be valuable. Future testing on datasets with different anatomical or textural characteristics could further strengthen the case for broader applicability, though this is not essential for the current manuscript.

Suggested Revision:

The authors may consider citing and briefly discussing known limitations of photogrammetry from external sources (e.g., relative to laser scanning or structured light), to strengthen the critical evaluation of their method's scope and boundaries.

Methodological Context

The manuscript presents a meaningful and well-documented improvement to photogrammetry workflows for biological specimen reconstruction. By integrating deep-learning-based masking (SAM) with structure-from-motion surface reconstruction (NodeODM) inside the 3D Slicer environment, the authors create a unified and accessible pipeline that reduces both user burden and reconstruction error.

The method is thoroughly validated through both geometric comparisons with micro-CT references and visual inspection of delicate anatomical regions. The use of real biological specimens (n = 14 skulls) strengthens the applicability of the results.

Although the comparison is primarily with their own previous workflow, the manuscript could be further strengthened by briefly situating this method within the broader landscape of available photogrammetry tools. This includes both commercial platforms and other open-source alternatives, often used in visualization and 3D model refinement. Doing so would highlight the distinct contribution of this workflow to the growing open-source infrastructure for morphology research.

Suggested Revision:

The authors may wish to **briefly** contextualize their method among other available solutions, including both commercial and open-source tools, to better highlight its unique advantages in usability, automation, or integration.

Minor Points / Suggestions:

1. Typos / Grammar:

- Abstract line 27: “The our improved pipeline” → should be “Our improved pipeline”.
- Several instances of slightly awkward phrasing (e.g., “clogs in narrow passages,” “bridging across suture lines”) – minor polishing would improve readability but isn't critical.

2. Terminology consistency:

- Terms like “ground truth” are used but could be better defined (i.e., clarify early that it refers to micro-CT models).
- “E → G” and “G → E” notation appears in Results without clear introduction – a short explanation upfront would help.

3. Figure clarity:

- If the actual figures match the descriptions (e.g., deviation maps, colored overlays), ensure that color maps are perceptually accessible, especially for red/green vision deficiencies, as required by BiO.
- Include scale bars where appropriate (e.g., on visual comparisons of skulls or maps).

4. Readability / Flow:

- The paper is dense but well-organized. Still, consider adding section headers or bullet points in the Results where performance metrics are explained (mean, RMSE, Hausdorff, Chamfer), to improve clarity for non-specialist readers.

Reviewer 2

Comments for the author

General Comments:

This study is about the development of a user-friendly open-source pipeline for processing photogrammetry data to produce higher resolution 3D models. This study builds on previous methods used by the authors, in their publication Zhang and Maga 2023. The authors have developed a Photogrammetry extension in the program 3D slicer, enabling the use of the Segment Anything Module to automate masking, and reconstructing surfaces using OpenDroneMap. This is a highly useful integrative and accessible tool that will be beneficial to the community to carry out morphological analyses for ecological and evolutionary research questions. To validate the improved performance of the new pipeline versus the previous one from Zhang and Maga 2023, the authors compare the resulting 3D models of 14 Mountain Beaver skulls (*Aplodontia rufa*) from both pipelines to microCT based 3D "ground truth" models. They show that mean distance and root mean square error (RMSE) are reduced in the new pipeline, indicating improved accuracy. However, Hausdorff and Chamfer metrics indicated mixed results, demonstrating the new model still requires improvements. While overall this study had merit for boosting efficiency and accuracy of capturing 3D data, the methods section lacks details that impacts its reproducibility. The resulting data could also be improved with further quantification, such as the completeness of fine structures like the zygomatic arches. Furthermore, there was little discussion between the new pipeline, compared to other methods outside the Zhang and Maga 2023 study highlighted. These limitations can certainly be fixed and would greatly improve the manuscript, particularly reproducibility which is crucial for a methods paper.

BiO Rubric Comments:

Experimental quality

This study could be further enhanced if smaller size specimens, such as mice skulls, were used to demonstrate the power of the new photogrammetry pipeline, and to check for the limitations of the resolution of this technique, but it's understandable this may be beyond the scope of this study. Independent or paired t-tests should be used to calculate significant difference between mean distance, RMSE, Hausdorff and Chamfer metrics between the two methods used.

It is explained in this study that Hausdorff distance score are affected by incompleteness of photogrammetry 3D models due to insufficient photo coverage. This suggests that perhaps Hausdorff distance scores are not a suitable measure to use to test the accuracy of this pipeline compared to microCT data. Other options that could improve the study are to place landmarks or point clouds on each mesh and calculate Procrustes Anova for analysis of difference. This was done in Zhang and Maga 2023 and may be a better way to indicate the accuracy of the 3D models.

Reproducibility

The methods section is lacking in some detail that impacts its reproducibility. This includes parameters selected in the Segment Anything Module. Final resolutions of 3D models from the photogrammetry pipelines are not reported. Finally, all details of the microCT data are missing and should be included or cited. See specific comments below for more detail.

Scholarship

This study did well to compare the new pipeline to that of Zhang and Maga 2023. However, it would also benefit from further discussion and comparison to other studies that compared photogrammetry and microCT pipelines, or those that have used open source automated tools for photogrammetry processing. Some examples include Marcy et al. 2018 (Peer J) <https://peerj.com/articles/5032/>, or Leo et al. 2025 (STAR Protocols) <https://www.sciencedirect.com/science/article/pii/S2666166724007378>.

Section Specific Comments:

Page 1 Line 27

Minor typo "The our pipeline"

Page 2 Line 43

Add "virtually" or "digitally" dissect

Page 2 Lines 59-60

It's not clear which references are linked to which statement in this paragraph

Page 3 Line 72

A 0.1-0.2mm discrepancy may be more significant for smaller animals and less significant for larger animals. Small needs to be defined here. Beavers are relatively large rodents for example.

Page 3 Line 77

It would be valuable to add whether this method is also applicable to users who use surface scanners instead of camera photogrammetry.

Page 4 Lines 87 and 88

Please add a following sentence to briefly explain what SAM and Open Drone Map tools are.

Page 4 Line 100

MicroCT resolution not given

Page 4 Line 105

Add common name "mountain beaver" to associate with species name *Aplodontia rufa*

Page 5 Lines 108-110

Some quantification (measurements, landmarks) or close up imaging of fine structures, such as zygomatic arch, foramina, pterygoid processes etc. would provide stronger support for this study, as imaging these fine structures is one of the main aims and strengths of developing this pipeline.

Page 5 Lines 114-116

Again, similarly to comment above, including quantification, such as number of polygons, points per area, or millimeter/micron resolution would be valuable to demonstrate improvement of 3D models between the two methods.

Page 5 Lines 116-119

The Hausdorff distances are higher in the new workflow, and both Hausdorff and Chamfer distances vary. This is explained by authors that there are skulls that were incompletely photographed. However, if the same photogrammetry dataset was used for both methods, both 3D models would have the same "incompleteness", and so why aren't the resulting Hausdorff distances a true reflection of the difference from the microCT shape between the two methods?

Page 5 Line 128 to Page 6 Line 131

Similarly to the above comment, please explain further how the Hausdorff distances are reflective of localised deficits. These deficits are presumably the same between methods if the same dataset has been used.

Page 6 Lines 134-136

Close ups of these features mentioned are needed in Figure 3. It is difficult to see the difference of these fine structures.

Page 6 Lines 146-149

The natural variation within skull morphology is not given, so it is difficult to know what the natural variation for skull morphology is. Is there error/variation taken between specimens to define this?

Page 7 Line 156-160

The resulting Chamfer distances is attributed to unmatched geometry between microCT and photogrammetry. However, if the same photo set was used for the two methods, they should be similarly mismatched to the microCT, but the 3D Slicer pipeline should still yield lower (more accurate) values. Could this be explained further?

Page 7 Line 163

In the Utility for Ecological and Evolutionary Applications it would be useful to give specific comparisons of the resolution/speed of the 3D Slicer pipeline to those used in other ecological/evolutionary studies.

Page 8 Line 177-181

This conclusion is more to do with the collection of photogrammetry data, similarly to Zhang and Maga 2023 conclusions, whilst it should be about improving the reconstruction of this data using the new pipeline. Comparisons with other similar studies using automated methods for masking and stitching would also be helpful.

Page 8 Line 189-192

Photogrammetry can always be improved by taking more photographs. This statement seems out of place here because this study is more about the processing of the dataset, rather than capturing the data. More helpful would be a discussion about why the new pipeline is better, is the machine learning more accurate at masking? What could be done better methods-wise in the future?

Page 9 Line 199-206

Details are missing for SAM options that were used in this study. For example, which photogrammetry module was used (ViT base, large or huge), what mask resolution (quarter, half). The parameters selected will impact the resulting resolution of the 3D models.

Page 9 Line 210

The MorphoCloud option should be included here with the Linux option, as it highlights that there are multiple platforms this pipeline can be operated in.

Page 11 Line 252-253

Which masking resolution was used for this study?

Page 15 Line 311

The use of the Taguchi L16 design is a clever way to determine optimal reconstruction settings. This should perhaps be mentioned in the introduction/results as a highlight of this pipeline/study.

Page 16 Line 327

There is no mention of where the microCT data is from, what instrument was used, what settings were used, the resolution achieved, and any processing/filtering of scan files afterwards. These can all impact the final resolution of the 3D models produced.

Page 23, Figure 2 needs to be checked for colour-blindness' readability as the red may not show up for red-green blindness.

Page 25, Figure 3 colours of boxplots are not colour-blind readable for people with monochromacy or if manuscript is printed in grayscale.

Page 29, Table 1 is missing UWBM in Specimen ID column

Reviewer's Responses to Questions

Experimental quality

Does each figure have the proper controls?

If 'No', please indicate reasons in Comments for Author box below.

Reviewer #1:

- Yes

Reviewer #2:

- No

Were the data analyzed using appropriate statistical tests?

If 'No', please indicate reasons in Comments for Author box below.

Reviewer #1:

- Yes

Reviewer #2:

- Yes

Reproducibility

Were experiments performed using adequate number of biological replicates?
If 'No', please indicate reasons in Comments for Author box below.

Reviewer #1:

- Yes

Reviewer #2:

- No

Does the methods section provide sufficient detail to permit reproducibility?
If 'No', please indicate reasons in Comments for Author box below.

Reviewer #1:

- Yes

Reviewer #2:

- Yes

Completeness

Are the manuscript's conclusions supported by the data?
If 'No', please indicate reasons in Comments for Author box below.

Reviewer #1:

- Yes

Reviewer #2:

- No

Scholarship

Do the authors cite and discuss the merits of data that would argue for and against their conclusion?
If 'No', please indicate reasons in Comments for Author box below.

Reviewer #1:

- Yes

Reviewer #2:

- No

Does the manuscript title & abstract accurately reflect the contents of the manuscript, without hyperbole?

If 'No', please indicate reasons in Comments for Author box below.

Reviewer #1:

- Yes

Reviewer #2:

- No

First revision

Author response to reviewers' comments

We thank the Editor and both reviewers for their constructive feedback. Below we reproduce every comment and provide a point-by-point response. Text that has been **added** to the manuscript is shown in *italic* or **bold** as appropriate, and line numbers refer to the marked-up revision.

1. Reviewer 1 - Experimental quality (general comment): “This study could be further enhanced if smaller size specimens, such as mice skulls, were used to demonstrate the power of the new photogrammetry pipeline...”

- Thank you for this thoughtful suggestion. We fully agree that evaluating performance across a range of specimen sizes—especially very small skulls—will be valuable for the community. In the present paper, however, our primary objective was to isolate the effect of the new masking and reconstruction steps by holding all other variables constant, including specimen identity, imaging conditions, and ground-truth micro-CT references. Re-shooting and re-scanning a second dataset of much smaller specimens (e.g., mouse skulls) would introduce additional confounding factors (different optics, lighting, depth-of-field constraints, and scale-dependent masking parameters) and would shift the scope from a controlled pipeline comparison to a broader performance survey.

2. Reviewer 1 - Statistics: “Independent or paired t-tests should be used to calculate significant difference between mean distance, RMSE, Hausdorff and Chamfer metrics between the two methods used.”

- [LN 120-125] Added paired t-test (n = 14) results for all four unsigned metrics (and for signed mean/median) in Results “Overview of Model Quality” and Table 1 footnote.

3. Reviewer 1 - Metric suitability: “Hausdorff distance scores are not a suitable measure... Other options... calculate Procrustes Anova for analysis of difference.”

- We appreciate the reviewer’s suggestion and have carefully considered alternative landmark-based statistics. However:

- i. Hausdorff distance is a field-standard metric for meshes. It directly captures the maximum point-to-surface deviation between two meshes and has been used for quantitative model-to-ground-truth comparisons
- ii. Dense semi- or pseudo-landmarks require a reliable point-to-point correspondence map between the photogrammetric and micro-CT meshes. Establishing that correspondence is non-trivial and that each algorithmic choice can alter statistical outcomes. These extra alignment steps risk inflating—or obscuring—the very error we seek to measure.
- iii. And perhaps more importantly, strictly LM based procrustes Anova would evaluate the difference at the anatomical landmarks, and would tell us nothing about the geometric accuracy of the model.
- iv. For these reasons, we have retained Hausdorff distance but (i) now clarify its role and limitations in the revised Methods, and (ii) commit to providing signed deviation maps in the supplementary material, as recommended by Reviewer 2.

4. Reviewer 1 - Reproducibility #1: “The methods section is lacking in some detail... This includes parameters selected in the Segment Anything Module.”

- a. [LN 324-340] Added “Architecturally, SAM’s encoder is a Vision Transformer (ViT-B, 12 layers, 768-dim embedding, 12-head self-attention) ...” to the “Image Acquisition and Masking” sub-section of the Methods and Materials.

5. Reviewer 1 - Reproducibility #2: “Final resolutions of 3D models from the photogrammetry pipelines are not reported.”

- a. Unlike grid-like volumetric datasets (e.g., microCT, CT, MR), surface models do not have a fixed “resolution”. For simple 3D geometry like a cube, a surface model with 12 triangles (2 triangles per face of cube) vs 12,000,000 will represent the same geometry with no loss of geometric accuracy. But if the reviewer is actually asking for descriptive information of model generation parameters and how consistent they were across, we now report for every reconstructed specimen:
 - i. **Vertex count and triangle count** (an objective measure of surface sampling density).
 - ii. **Texture map resolution**: all textures were generated at the maximum size permitted by NodeODM to avoid down-sampling.
- b. These statistics are provided in the revised Supplementary Table 1, allowing readers to gauge the effective surface sampling resolution and compare it directly between the legacy and new workflows.

6. Reviewer 1 - Reproducibility #3: “All details of the microCT data are missing and should be included or cited.”

- a. [LN 347-357] Added “Micro-CT Reference Imaging and Model Extraction” to the methods section.

7. Reviewer 1 - Scholarship broader context: “It would also benefit from further discussion and comparison to other studies... Marcy et al. 2018... Leo et al. 2025...”

- a. [LN 198-213] Added “Comparison with alternative 3-D digitisation technologies” subsection to the Results/Discussion.

8. Reviewer 1 - p. 1 L27: “Minor typo ‘The our pipeline’”

- a. [LN 29-30] Corrected.

9. Reviewer 1 - p. 2 L43: “Add ‘virtually’ or ‘digitally’ dissect”

- a. [LN 43] Added “digitally” to the sentence as suggested.

10. Reviewer 1 - p. 2 L59-60: “It’s not clear which references are linked to which statement in this paragraph”.

- a. [LN 57-60] The references have been separated for clarity.

11. Reviewer 1 - p. 3 L72: “Small needs to be defined here.”

- a. [LN 71-76] Added “Dashti et al., 2022 suggest that discrepancies of 0.1-0.2 mm—approximately 0.25-0.5 % of condylobasal length in the *Tatera indica* skulls

measured by Dashti et al. (2022; CBL \approx 38-44 mm)—have already been shown to obscure population-level shape signals. Because the mountain-beaver skulls examined here are larger, the same absolute error represents an even smaller proportion of total length, yet still matters for fine-scale morphometrics.”

12. Reviewer 1 - p. 3 L77: “Add whether this method is also applicable to users who use surface scanners instead of camera photogrammetry.”

- a. [LN 82-84] Added “It does not focus on reconstructing 3D models based on data obtained from laser surface scanners or other related technology.”

13. Reviewer 1 - p. 4 L87-88: “Please add the following sentence to briefly explain what SAM and Open Drone Map tools are.”

- a. [LN 91-107] Added “Here, we improve on the previous study by integrating the Segment Anything Model (SAM) (Kirillov et al., 2023) with structure-from-motion reconstruction (NodeODM, part of the OpenDroneMap project; Patel et al., 2024) and combining these tools within the 3D Slicer environment (Fedorov et al., 2012) to unify data acquisition, segmentation, reconstruction, and final model generation.”

14. Reviewer 1 - p. 4 L100: “MicroCT resolution not given”

- a. [LN 347-357] Added “Micro-CT Reference Imaging and Model Extraction” to the methods section.

15. Reviewer 1 - p. 4 L105: “Add common name ‘mountain beaver’”

- a. [LN 25, 218] Done in Abstract and Results/Discussion.

16. Reviewer 1 - p. 5 L108-110: “Some quantification... close-up imaging of fine structures... would provide stronger support.”

- a. Added Supplementary Figure 1, close-up views of fine structures for original and new photogrammetry workflows.

17. Reviewer 1 - p. 5 L114-116: “Including quantification, such as number of polygons, points per area, or millimeter/micron resolution would be valuable...”

- a. Added Supplementary Table 1 with model vertex and triangle counts, and texture resolutions.

18. Reviewer 1 - p. 5 L116-119: “If the same photogrammetry dataset was used for both methods... why aren’t the resulting Hausdorff distances a true reflection...?”

- a. We agree that both workflows start from an identical photo set; however, the geometry each pipeline reconstructs from those images is **not identical** (which is the important point of our current study), and the Hausdorff distance is extremely sensitive to any local discrepancy—even a single out-of-place vertex can dominate the metric.

19. Reviewer 1 - p. 5 L128-p. 6 L131: “Please explain further how the Hausdorff distances are reflective of localised deficits.”

- a. [LN 176-181] Added “Because Hausdorff is defined by the single largest vertex-to-vertex deviation...” to the “Metrics and Their Interpretations” section of the Results/Discussion.

20. Reviewer 1 - p. 6 L134-136: “Close ups of these features mentioned are needed in Figure 3.”

- a. We’ve added a new Supplementary Figure 1, which shows close-up views of these structures on a single specimen.

21. Reviewer 1 - p. 6 L146-149: “The natural variation within skull morphology is not given...”

- a. Because the two reconstructions being compared originate from exactly the same set of photographs of the same individual skulls, population-level morphological variation does not enter into our error analysis. Any discrepancies we measure must therefore arise from methodological differences (masking strategy, MVS

parameters, model generation parameters) rather than from biological variation among specimens.

22. Reviewer 1 - p. 7 L156-160: "Could this be explained further?" (re: Chamfer distances)

- a. [LN 185-191] Added "[b]y averaging the nearest-neighbour distances in both directions across all vertices, Chamfer yields an outlier-resistant, global measure of surface correspondence that complements the extremal sensitivity of Hausdorff distance."

23. Reviewer 1 - p. 7 L163: "Give specific comparisons of the resolution/speed of the 3D Slicer pipeline to those used in other ecological/evolutionary studies."

- a. Added mesh statistics (including resolution) and reconstruction times for each reconstructed model to Supplementary Table 1.

24. Reviewer 1 - p. 8 L177-181: "This conclusion... should be about improving the reconstruction... Comparisons with other similar studies... would also be helpful."

- a. [LN 198-213] Added "Comparison with alternative 3-D digitisation technologies" subsection to the Results/Discussion. But beyond the discussion, we cannot conduct direct comparison with other studies, because most (if not all up to this point) studies of photogrammetry use proprietary software that we do not have access to. Hopefully researchers who have access to these software can indeed compare open-source and proprietary software since we provide all of our raw and processed datasets in future, stimulating new discussions in the field.

25. Reviewer 1 - p. 8 L189-192: "More helpful would be a discussion about why the new pipeline is better... What could be done better method-wise in the future?"

- a. [LN 241-249] Added "[t]he observed gains stem from three synergistic upgrades: (i) SAM preserves crisp object boundaries under low contrast, dramatically reducing background leakage; (ii) NodeODM's multi-view-stereo stage recovers high-frequency surface detail that was previously lost to coarse Poisson smoothing; and (iii) our Taguchi-driven parameter search selects near-optimal reconstructions without manual trial-and-error. Future iterations will explore depth-map fusion for texture-poor areas and adaptive mask refinement guided by photometric consistency checks—improvements we expect to yield further sub-millimetre accuracy, particularly for smaller or more weakly textured specimens.

26. Reviewer 1 - p. 9 L199-206: "Details are missing for SAM options... which photogrammetry module was used... what mask resolution..."

- a. [LN 324-336] Added "[a]rchitecturally, SAM's encoder is a Vision Transformer (ViT-B, 12 layers, 768-dim embedding, 12-head self-attention) that tokenises each 1024×1024 image into 16×16 -pixel patches and augments them with learned positional embeddings. The global [CLS] token aggregates context across all patches, and the resulting latent features are passed to SAM's lightweight mask-decoder for prompt-conditioned segmentation (Kirillov et al., 2023). The extension loads the official Segment Anything "ViT-B" checkpoint by default and executes inference on the full-resolution RGB images on GPU if available (otherwise CPU). The UI lets users down-sample the display to full, $\frac{1}{2}$, or $\frac{1}{4}$ size for quicker interaction, but the predictor itself always receives the native-resolution images, and it returns a single mask per bounding box."

27. Reviewer 1 - p. 9 L210: "The MorphoCloud option should be included here with the Linux option..."

- a. [LN 276-277] Added "...and via MorphoCloud On-Demand (<https://github.com/MorphoCloud>) if a cloud-based approach requiring no installation or setup is preferred."

28. Reviewer 1 - p. 11 L252-253: "Which masking resolution was used for this study?"

- a. [LN 338-340] Added "[w]e retained the maximum masking resolution for every reconstruction because our workstation's computing resources were sufficient to handle the added load without down-sampling."

29. **Reviewer 1 - p. 15 L311: “The use of the Taguchi L16 design... should perhaps be mentioned in the introduction/results as a highlight.”**
- [LN 91-107] Added “Here, we improve on this study by integrating the Segment Anything Model (SAM) (Kirillov et al., 2023) with structure-from-motion reconstruction (NodeODM...” to the last portion of the Introduction.
30. **Reviewer 1 - p. 16 L327: “There is no mention of where the microCT data is from, what instrument was used, what settings were used...”**
- [LN 347-357] Added “Micro-CT Reference Imaging and Mesh Extraction” to the methods section.
31. **Reviewer 1 - Figure 2: “Needs to be checked for colour-blindness’ readability...”**
- This figure has been updated.
32. **Reviewer 1 - Figure 3: “Colours of boxplots are not colour-blind readable...”**
- This figure has been updated.
33. **Reviewer 1 - Table 1: “Table 1 is missing UWBM in Specimen ID column.”**
- The full specimen IDs have been added to Table 1.
34. **Second reviewer document - Revision #1: “The authors may consider including a signed distance metric... which would complement the existing evaluation.”**
- I’ve updated Figure 3 and the Supplementary Figures for each specimen with signed distance representations of error.
35. **Second reviewer document - Reproducibility request #1: “Please consider completing the example dataset (UWBM 82409) by adding the associated masks, reconstructed model, and evaluation output files...”**
- I’ve included a new link to these additional files.
36. **Second reviewer document - Reproducibility request #2: “Sharing at least 1-2 more specimens... would help demonstrate consistency...”**
37. **Second reviewer document - Reproducibility request #3: “Consider clarifying the long-term availability and platform support for users on Windows/macOS who cannot use MorphoCloud or prefer local workflows.”**
- [LN 311-316] The installation paragraph now states: “While the extension isn’t available for Windows and MacOS users via the 3D Slicer Extensions Manager, users can directly download the extension from GitHub, manually install the pre-requisites, and then use the 3D Slicer Extension Wizard.”
38. **Second reviewer document - Conclusions: “Highlighting the limitations more explicitly in the conclusions section...”**
- [LN 256-254] Added a new section to the conclusions “While the new workflow narrows photogrammetry error to ~0.3 mm, several inherent limitations remain: (i) only external geometry is captured, so internal cavities or trabecular networks must still be imaged by micro-CT; (ii) accuracy degrades sharply when image coverage is incomplete, surface texture is low, or specimens are <10 mm, and (iii) highly specular or translucent materials can confound both SAM masking and SfM matching...”
39. **Second reviewer document - Scholarship broader context: “Briefly situating this method within the broader landscape of available photogrammetry tools...”**
- [LN 198-213] Added “Comparison with alternative 3-D digitisation technologies” subsection to the Results/Discussion.
40. **Second reviewer document - Minor typo: “Abstract line 27: ‘The our improved pipeline’ → should be ‘Our improved pipeline’.”**
- Fixed.

41. **Second reviewer document - Minor phrasing:** “Several instances of slightly awkward phrasing... minor polishing would improve readability.”
 42. **Second reviewer document - Terminology:** “Terms like ‘ground truth’ are used but could be better defined...”
 - a. [LN 111-114] Added “A high-resolution micro-CT scan of the specimens serves as the ground-truth reference, against which photogrammetric models produced by the legacy and updated workflows—using the same image set—are quantitatively compared.”
 43. **Second reviewer document - Notation:** “‘E → G’ and ‘G → E’ notation appears... without clear introduction.”
 - a. [LN 143-145] Added the sentence “Here, E denotes the photogrammetry-derived (E)xperimental model and G the micro-CT (G)round-truth surface (the external surface of the CT mesh); unless noted otherwise, distances are reported from E → G” to the “Comparisons Between Old and New Workflows” section of the results.
 44. **Second reviewer document - Figure accessibility:** “Ensure that color maps are perceptually accessible... Include scale bars where appropriate.”
 - a. All Figures are now colorblind accessible, and scalebars have been added to Figure 3.
-

Second decision letter

MS ID#: bio.062126R1

MS Title: SlicerMorph Photogrammetry: An Open-Source Photogrammetry Workflow for Reconstructing 3D Models

Authors: Oshane O. Thomas, Chi Zhang and A. Murat Maga

I've had the opportunity to thoroughly read your rebuttal and the associated changes you've made to your manuscript. I believe you've addressed the criticisms well. Consequently, I am happy to tell you that your manuscript has been accepted for publication in Biology Open, pending our standard publication integrity checks. It was accepted on 25 July 2025.